# Combination Therapy and Phytochemical-Loaded Nanosytems for the Treatment of Neglected Tropical Diseases

**DOI:** 10.3390/pharmaceutics16101239

**Published:** 2024-09-24

**Authors:** Jacqueline Soto-Sánchez, Gilberto Garza-Treviño

**Affiliations:** Section for Postgraduate Studies and Research, Escuela Nacional de Medicina y Homeopatía, Instituto Politécnico Nacional, Guillermo Massieu Helguera #239, Fracc. La Escalera, Ticomán, Ciudad de México 07320, Mexico; ggarzat@ipn.mx

**Keywords:** phytochemicals, leishmaniasis, trypanosomiasis, schistosomiasis, combination therapy, nanosystems

## Abstract

**Background:** Neglected tropical diseases (NTDs), including leishmaniasis, trypanosomiasis, and schistosomiasis, impose a significant public health burden, especially in developing countries. Despite control efforts, treatment remains challenging due to drug resistance and lack of effective therapies. **Objective**: This study aimed to synthesize the current research on the combination therapy and phytochemical-loaded nanosystems, which have emerged as promising strategies to enhance treatment efficacy and safety. **Methods/Results**: In the present review, we conducted a systematic search of the literature and identified several phytochemicals that have been employed in this way, with the notable efficacy of reducing the parasite load in the liver and spleen in cases of visceral *leishmaniasis*, as well as lesion size in cutaneous *leishmaniasis*. Furthermore, they have a synergistic effect against *Trypanosoma brucei rhodesiense* rhodesain; reduce inflammation, parasitic load in the myocardium, cardiac hypertrophy, and IL-15 production in *Chagas disease*; and affect both mature and immature stages of *Schistosoma mansoni*, resulting in improved outcomes compared to the administration of phytochemicals alone or with conventional drugs. Moreover, the majority of the combinations studied demonstrated enhanced solubility, efficacy, and selectivity, as well as increased immune response and reduced cytotoxicity. **Conclusions**: These formulations appear to offer significant therapeutic benefits, although further research is required to validate their clinical efficacy in humans and their potential to improve treatment outcomes in affected populations.

## 1. Introduction

The despite the advent of novel anti-parasitic therapies, neglected tropical diseases (NTDs), which are prevalent in populations in tropical and subtropical countries [1], continue to represent a significant global health challenge. These NTDs encompass diseases caused by protozoan pathogens, including leishmaniasis and sleeping sickness, Chagas disease, as well as helminthic pathogens such as schistosomiasis. The persistence of these diseases can be attributed to the evolution of sophisticated strategies employed by parasites that allow them to evade and modulate host immune responses, coupled with high fecundity that ensures the perpetuation of their invasive forms [2,3,4]. Despite the use of therapeutic agents, their efficacy is limited, and they can induce severe or even lethal side effects (Table 1) [5]. Antiparasitic drugs encounter considerable obstacles when attempting to treat intracellular parasites, mainly due to their limited penetration through biological membranes. This results in reduced bioavailability and diminished therapeutic efficacy [6]. In the case of leishmaniasis and Chagas disease treatment, this poses a further challenge, as the drugs must accumulate in the phagolysosomal compartment of host macrophages and selectively kill the parasite without cytotoxic effects [7,8]. Among the strategies proposed for the identification of new antiparasitic compounds, drug combinations have been identified as a promising approach to reducing the required effective concentration and improving therapeutic efficacy. The combination of multiple drugs, which act on different targets or signaling pathways, results in a notable reduction in the concentrations required for each compound. This not only enhances efficacy but also mitigates toxicity and reduces the likelihood of developing resistance to treatments [9].

On the other hand, an alternative approach to the search for new antiparasitics is to focus on phytochemicals derived from natural sources. These include polyphenols, flavonoids, lignans, chalcones, alkaloids, and triterpenes, which represent promising antiparasitic alternatives. These bioactive molecules have been demonstrated to possess leishmanicidal, trypanosomicidal, and schistosomicidal properties and to target different biochemical targets, thereby opening up a promising avenue for the exploration of new treatment modalities with potentially fewer side effects and greater efficacy [34,35,36]. Nevertheless, the physicochemical and pharmacokinetic properties of bioactive compounds markedly impede their therapeutic efficacy. Firstly, the low water solubility and limited stability of these compounds in the gastrointestinal tract, which is affected by several factors including enzymes, pH, microbiota, and the presence of other nutrients, play a significant role in determining their absorption [37,38]. Furthermore, the water solubility of these secondary metabolites is influenced by their affinity for water molecules and their molecular weight (MW), such that higher MW is associated with lower solubility [39]. For example, resveratrol (Rsv) exhibits high solubility in alcohol but low solubility in water (21 μg/mL at pH 7.4), which constrains its bioavailability [40]. Similarly, betulinic acid (BA) has an extremely low solubility in water (0.02 μg/mL at room temperature), and its solubility in organic solvents is also limited [41]. Furthermore, additional factors influencing the bioavailability of these bioactive molecules include high metabolic rates, inactivation of metabolic products, and rapid elimination and clearance [42]. Consequently, the poor bioavailability of these phytochemicals severely restricts their potential as antiparasitic agents. To overcome the aforementioned limitations, nanosystems such as nanostructured lipid carriers (NLCs), nanoemulsions (NEs), polymeric nanoparticles (PNPs), and metallic nanoparticles (MNPs) have been employed. These systems can facilitate the protection of the drug against degradation in physiological media, enhance drug solubility, and modify the drug, thereby enabling its transport across biological membranes [43,44,45]. Accordingly, this review aims to delineate the principal contributions to combination therapy with phytochemicals and nanotechnologies over the past five years, with a particular focus on the delivery of secondary metabolites. Furthermore, it will examine and analyze the enhancement of the leishmanicidal, trypanosomicidal, and schistosomicidal properties of these phytopharmaceuticals in accordance with their nanostructured vehicles. The mechanisms of action, synergistic effects, and therapeutic advantages of the use of phytochemical-loaded nanosystems for the treatment of these NCDs will also be examined and analyzed.

## 2. Combination Therapy with Phytochemicals for the Treatment of Visceral Leishmaniasis

Leishmaniasis, a parasitic disease caused by the protozoan *Leishmania*, affects cells of the phagocytic mononuclear system and is transmitted to humans through the bite of infected female sandflies. The disease is endemic in 99 countries and threatens more than 350 million people worldwide, causing approximately 50,000 deaths annually. Leishmaniasis manifests in four different clinical forms: visceral, cutaneous (diffuse and localized), mucocutaneous, and dermal post-kala-azar. The parasite has two main forms in its life cycle: promastigotes, which develop extracellularly in the vector, and amastigotes, which develop intracellularly in the macrophages of the mammalian host. Within the host macrophages, *Leishmania* sp. inhibits nitric oxide (NO) generation and reduces the production of pro-inflammatory cytokines such as interleukin-12 (IL-12) and interferon-γ (IFN-γ), while increasing the production of anti-inflammatory cytokines such as IL-10 and transforming growth factor-beta (TGF-β) [46,47].

Visceral leishmaniasis (VL), caused by the parasites *L. donovani*, *L. infantum*, and *L. chagasi*, is commonly known as kala-azar disease. It represents the most severe form of the disease, affecting the reticuloendothelial cells of organs such as the spleen, liver, and bone marrow. The principal clinical manifestations are fever, hepatomegaly, and pancytopenia. The World Health Organization (WHO) estimates that 50,000–90,000 new cases of VL occur annually globally [48,49]. Given the severe adverse effects of the currently available leishmaniasis chemotherapy drugs and the emergence of parasitic resistance (Table 1), there is a pressing need to identify new therapeutic targets and strategies that are more effective, less toxic, and require shorter administration cycles. In the field of combination therapy research against VL, lupeol (Lup), ursolic acid (UA), and cynaroside (Cye) (Figure 1) are emerging as natural compounds with distinctive properties and promising therapeutic potential. Lup is a pentacyclic triterpene with excellent in vitro and in vivo activity against VL [35,50]. Studies in BALB/c mice infected with L. donovani have revealed that the suboptimal combination of Lup with amphotericin B (AmpB) not only significantly reduces the parasite load in the liver and spleen but also enhances the effectiveness of AmpB (Table 2). This is because the Lup–AmpB combination increases the production of Th1 cytokines (IL-12 and IFN-γ) while suppressing the production of Th2 cytokines (IL-10 and TGF-β), which increases the observed NO production [51].

Another pentacyclic triterpene is UA, which is present in several medicinal plants and exhibits activity against *L. infantum* [58]. Studies in mice showed that the UA–AmpB combination reduces hepatic and splenic parasitism levels more effectively than AmpB monotherapy on *L. infantum* promastigotes (Table 2). Although this combination had no significant impact on IFN-γ and IL-10 expression [52], UA has been shown to affect several enzymes crucial for *L. donovani* metabolism, such as pteridine reductase 1, farnesyl diphosphate synthase, and N-myristoyl transferase [59]. Another active metabolite is Cye, a flavonoid found in many plants. This compound exhibited in vitro activity on *L. donovani* promastigotes lower than that of miltefosine (MTF). However, the MTF–Cye combination inhibited the growth of the parasites by more than 50% and altered their morphology by decreasing their size and changing their shape, while in intramacrophage amastigotes, the effect was even more potent, reducing the number of amastigotes by up to 80%. The mechanism of action may be related to increased production of ROS and inhibition of cell cycle progression in *L. donovani* promastigotes. Interestingly, Cye inhibits in silico UDP-galactopyranose mutase (UGM), which is a flavin-dependent enzyme key in the biosynthesis of UDP-galactofuranose (UDP-Galf), which acts as a precursor for the synthesis of β-galactofuranose (β-Galf), which is an important component of the *Leishmania* cell surface matrix and plays a key role in the pathogenesis of the parasite [53,60].

### 2.1. Combination Therapy with Phytochemicals for the Treatment of Cutaneous Leishmaniasis

Cutaneous leishmaniasis (CL), caused by species such as *L. major*, *L. tropica*, *L. mexicana*, and *L. amazonensis*, represents the most prevalent form of this disease globally, with approximately 1 million cases reported annually, according to the WHO. This pathology causes disfiguring lesions on exposed skin, generating considerable social stigma and significant disability [49]. Conventional treatment carries risks of serious side effects (Table 1). In vitro studies have shown that crocin (Crn) (Figure 1), a tetraterpenic carotenoid pigment, exhibits a leishmanicidal effect, especially when combined with AmpB (Crn-AmpB), showing potent synergy against intracellular amastigotes. This combination has also demonstrated a less pronounced effect on *L. major* promastigotes, being less toxic due to an increased selectivity index (SI) compared to AmpB monotherapy (Table 2). The underlying mechanism involves the inhibition of arginase (L-ARG), an enzyme crucial for parasite survival, in addition to enhancing the Th1 response, blocking the cell cycle, and promoting apoptosis [54]. In an animal model of CL caused by *L. amazonensis*, treatment with glucantime (Glu) as monotherapy showed limited activity in dermal parasite clearance. However, the combination of ursolic acid and glucantamine (UA-Glu) significantly reduced the number of viable parasites on the skin by 99.8% (Table 2) and showed a trend toward healing skin lesions superior to that of monotherapy treatments. This effect was associated with lesion improvement and a positive up-regulation in IFN-γ production, as well as a decrease in IL-4 production [52]. In contrast, combinations such as limonene–carvacrol (Lim-Car) demonstrated no remarkable activity against *L. major* amastigotes and promastigotes [55], and combinations such as Glu–gamma-terpinene (GT) and farnesol (FA) + paromomycin (Par) proved to be antagonistic (Table 2), possibly due to the low solubility of these bioactive compounds [56,57].

### 2.2. Phytochemical-Loaded Nanosystems for the Treatment of Visceral Leishmaniasis Caused by L. infantum

Lupeol (Lup) and ursolic acid (UA) are two terpenes that have been demonstrated to exhibit excellent activity against VL. However, their low water solubility (log Sw = −8.57 and −7.23) and intermediate lipophilicity (Log P = 7.6 and 5.93) [61] present significant challenges in their therapeutic application. To overcome these disadvantages, these compounds can be encapsulated in different types of nanocarriers, including nanostructured lipid carriers (NLCs). NLCs are a binary system containing solid and liquid lipids, which in turn produce a less ordered lipid core. This imperfect internal arrangement leads to greater drug accommodation, which offers significant advantages, such as zeta potential (ZP) stability, long-term particle size (PS) stability, high drug-loading capacity, and prevention of systemic adverse effects [62]. In a trial in which Lup-loaded nanostructured lipid carriers (Lup-NLCs) were administered to *L. infantum*-infected golden hamsters, a greater reduction in the number of parasites in the spleen and liver was observed compared to the untreated infected control and Lup-treated animals. The results observed in animals treated with AmpB were comparable to those observed in animals treated with Lup-NLCs (Table 3). Furthermore, animals treated with Lup-NLCs exhibited enhanced preservation of the spleen and liver, discrete macrophage hyperplasia in the spleen—suggestive of a protective effect against disease progression—and elevated levels of anti-*Leishmania* IgG2 antibodies, indicative of a developed Th1 immune response [63]. The administration of UA-loaded nanostructured lipid carriers (UA-NLCs) was observed to result in a notable reduction in parasite load within the spleen and liver of *L. infantum*-infected hamsters in comparison to the administration of free UA and AmpB (Table 3). In addition, UA-NLCs were found to modulate the immune response, increasing IFN-γ expression and promoting IgG2 production, which is associated with a Th1-type immune response (Figure 2). In contrast to the kidney-damaging effects of AmpB administration, the UA-NLCs formulation was reported to be non-toxic [64]. It has also been demonstrated that artemisinin-loaded solid lipid nanoparticles (Ar-SLNs) (Figure 3) are efficacious in the treatment of experimental VL. In a murine model of *L. infantum* infection, treatment with Ar-SLNs resulted in a notable reduction in parasite burden within the liver and spleen when compared to free Ar (Table 3). However, this effect was less pronounced than that observed with the combination of Lup-NLCs and UA-NLCs [65].

### 2.3. Phytochemical-Loaded Nanosystems for the Treatment of Visceral Leishmaniasis Caused by L. donovani

The presence of hydroxy (-OH) and carbonyl (-CO) groups, in conjunction with aromatic rings, is mainly responsible for the high reducing potential of flavonoids; this property is useful in the preparation of nanoformulations as it allows the reduction of metal ions and stabilizes the structure and stability of the nanoparticles. The interaction of biomolecules with metal ions leads to the formation of conjugated nanoparticles, which have a higher efficiency than the free biomolecule [69]. 4′,7-dihydroxyflavone (4′,7-DHF) (Figure 3) is a natural flavonoid compound found in various plants with low solubility (Log Po/w = 2.40 and LogS = −4.03) and has shown remarkable activity against *L. donovani* promastigotes and amastigotes [67]. To improve its selectivity and efficacy, 4′,7-dihydroxyflavone was conjugated to multi-walled carbon nanotubes (4′,7-DHF-MWCNTs), which are characterized by their excellent ability to penetrate cell membranes and be readily phagocytosed by macrophages [74,75] as well as gold and silver nanoparticles (4′,7-DHF-AuNPs and 4′,7-DHF-AgNPs). It has been observed that both AgNPs and AuNPs exhibit the ability to generate ROS and are readily phagocytosed by macrophages, which are the main site of proliferation and growth of the *Leishmania* parasite [76,77]. The three formulations were evaluated against *L. donovani* promastigotes, axenic amastigotes, and intracellular amastigotes. The 4′,7-DHF-MWCNT formulation showed approximately 28-, 38-, and 11-fold higher activity compared to 4′,7-DHF administration alone, whereas the 4′,7-DHF-AuNPs and 4′,7-DHF-AgNPs formulations exhibited approximately 25- and 12-fold higher activity against axenic amastigotes, respectively, than administration of 4′,7-DHF alone. Interestingly, conjugation of 4′,7-DHF with MWCNTs, AuNPs, and AgNPs was observed to increase its cytotoxicity and decrease its SI (Table 3). The mechanism of action of both 4′,7-DHF and its three conjugations appears to be similar, by inducing high levels of ROS, which ultimately leads to parasite death [66,67,68]. In this context, the compound 4′,7-DHF is also known to target the *L. donovani* enzyme tyrosine aminotransferase. This enzyme is very important for the survival of the parasite inside the macrophage, as it is involved in the scavenging of ROS [78]. Chrysin (5,7-dihydroxyflavone) (Chy) (Figure 3) is another flavone containing two hydroxy (–OH) groups at positions 5 and 7 of the A-ring. Major natural sources of Chy include passion fruit *(Passiflora* sp.), honey, and propolis [79]. Recent findings also revealed that Chy specifically targets mitogen-activated protein kinases (MPKs), MPK3, and MPK4 of *L. donovani*. Ld MAPK3 is implicated in flagella length regulation and thus plays an essential role in disease transmission, while Ld MAPK4 is crucial for parasite survival through its involvement in various regulatory, apoptotic, and developmental pathways [80,81]. However, Chy has poor solubility and also shows low bioavailability due to its rapid metabolism in the gastrointestinal tract [82]. Therefore, to increase the efficiency and solubility, the reductive property of Chy was employed to reduce metal ions, with the resulting compound functionalized on AuNPs. Fourier transform infrared (FTIR) spectroscopy analysis confirmed the active participation of the hydroxyl (–OH) and carbonyl (C=O) groups of Chy while forming Chy-conjugated AuNPs (Chy-AuNPs). The findings demonstrated that Chy-AuNPs exhibited ~3 times greater efficacy than Chy against *L. donovani* intracellular amastigotes (Table 3) [69]. On the other hand, quercetin (QT) (Figure 3) is a natural flavonol with a wide range of biological activities, including activity against various parasites such as *Trypanosoma* spp., as well as against several species of *Leishmania*, such as *L. donovani* and *L. amazonensis* [83,84]. However, it has low aqueous solubility and poor oral bioavailability (~17%), which has limited its therapeutic efficacy and application [85]. A lipid-based delivery system is an optimal choice for the administration of hydrophobic bioactives such as QT, as these systems enhance the solubility and bioavailability of hydrophobic components [86]. Accordingly, Sashi et al. (2022) assessed the antileishmanial efficacy of quercetin-loaded nanoemulsion (QTNE) against clinical strains of *L. donovani*. The results of the in vitro anti-promastigote assay showed that QTNE exhibited similar efficacy to QT against *L. donovani* (Table 3) [70]. In addition, QTNE has been demonstrated to increase intracellular levels of ROS and NO and induce distortion of membrane integrity and release of phosphatidylserine (Figure 2) [70]. Another flavone with noteworthy activity against *L. donovani* is 7,8-dihydroxyflavone (7,8-DHF) (Figure 3). This compound decreases the in vitro growth of *L. donovani* axenic amastigotes with a half-maximal inhibitory concentration (IC_50_) of 1.7 µg/mL; however, it has low water solubility, chemical instability, and poor bioavailability due to glucuronidation and sulfation in vivo [73]. To enhance these characteristics and improve its efficacy, 7,8-DHF was conjugated to AuNPs (7,8-DHF-AuNPs). 7,8-DHF-AuNPs were observed to be more effective against promastigotes as well as against AmpB-resistant promastigotes and SAG (sodium antimony gluconate)-resistant promastigotes than free 7,8-DHF (Table 3). Treatment with 7,8-DHF-AuNPs inhibits arginase in parasites, which disrupts the polyamine biosynthesis pathway by converting *l*-arginine to ornithine. This, in turn, limits the production of putrescine, a crucial component for parasite survival. While these treatments do not affect the expression of downstream enzymes such as ornithine decarboxylase (ODC), they do significantly reduce the expression of the multidrug resistance gene (MDR1), a membrane transport protein associated with multidrug resistance of the parasite. Treatment with 7,8-DHF-AuNPs also increased Th1 response versus decreased Th2 response, leading to increased host nitric oxide synthase (*iNOS)* (Figure 2) activity and subsequent parasite clearance [72]. It is noteworthy that among the three flavones (4′,7-DHF, 5,7-DHF, 7,8-DHF) conjugated to AuNPs, 4′,7-DHF exhibited the most pronounced inhibitory activity against *L. donovani*. The enhanced activity against *Leishmania* may be attributed to the presence of an OH group on the B-ring of the flavone and its role in stabilizing the AuNPs. The efficacy of these conjugated nanoparticles should be further validated by in vivo assays and animal models.

### 2.4. Phytochemical-Loaded Nanosystems for the Treatment of Cutaneous Leishmaniasis Caused by L. amazonensis

*L. amazonensis* is one of the species that causes localized CL, a clinical variant characterized by one or a few lesions that initially manifest as papules in the area of the bite of an infected sandfly. These lesions gradually evolve into ulcers [87]. 4-Nitrochalcone (4NC) (Figure 3) is a naturally occurring flavonoid with two aromatic rings linked through a three-carbon enone moiety. It is insoluble in water. To increase its solubility and efficacy, 4-Nitrochalcone was loaded into beeswax-copaiba oil nanoparticles (4NC-beeswax-CONPs). The beeswax-CONPs are composed of beeswax, which has antimicrobial properties, and copaiba oil (*Copaifera* sp.), which demonstrated biological action on *L. amazonensis* promastigotes and amastigotes [88,89]. The leishmanicidal effect of 4NC on intracellular amastigotes and 4NC-beeswax-CONPs was found to be highly comparable. It is noteworthy that when free 4NC was tested in macrophages, it exhibited cytotoxicity (CC_50_ = 8.73 μM). This toxicity may be attributed to the NO₂ group conjugated to one of the aromatic rings, as the presence of this group has been previously associated with toxicity [90,91]. In contrast, 4NC-beeswax-CONPs did not alter the viability of peritoneal macrophages (CC_50_ > 50 μM), which led to an improvement in the SI with respect to that presented by 4NC (Table 4) [92]. In a previous study, it was observed that beeswax-CONPs loaded with diethyldithiocarbamate improved selectivity against parasites [88]. 4NC-beeswax-CONPs increased TNF-α production in macrophages infected with *L. amazonensis* (Figure 2). TNF-α is a proinflammatory cytokine that participates in macrophage activation and induces ROS and NO production. In the same study, it was also found that 4NC-beeswax-CONPs induced ROS and NO production, while free 4NC induced only ROS production [92].

### 2.5. Phytochemical-Loaded Nanosystems for Combating Cutaneous Leishmaniasis Caused by L. major

Betulinic acid (BA) and linalool (Li) (Figure 3) are two naturally occurring terpenes that exhibit activity against various *Leishmania* species [97,98], ranging from mild to moderate. However, their clinical use is limited by their low water solubility and short plasma half-life [99,100,101]. To overcome these limitations, chitosan nanoparticles (KNPs), known for their immunological properties [102], and zinc oxide nanoparticles (ZNPs), which have antileishmanial effects on *L. major* promastigotes, were developed [103]. BA-loaded chitosan nanoparticles (BA-KNPs) demonstrated significant lethal effects against *L. major* promastigotes and intracellular amastigotes in vitro, similar to amphotericin B-loaded chitosan nanoparticles (AmpB-KNPs) (Table 4). In vivo studies demonstrated that administration of BA-KNPs resulted in a complete reduction in lesion size in *L. major*-infected mice after 6 weeks post-treatment [6]. On the other hand, linalool-loaded zinc oxide nanoparticles (Li-ZNPs) showed superior activity against intracellular amastigotes and promastigotes *L. major* compared to Glu, a commonly used drug. Despite exhibiting slightly elevated toxicity relative to Glu, Li-ZNPs effectively diminished lesion size in mice following 28 days of treatment (Table 4) [93]. Previous studies have demonstrated the induction of apoptosis in host cells, which represents a crucial mechanism for the management and elimination of pathogens. Flow cytometry analysis revealed a notable increase in necrotic and apoptotic cells following Li-ZNPs treatment, indicating their potential antileishmanial activity through the induction of apoptosis. Additionally, the positive regulation of gene expression levels of iNOS, IFN-γ, and TNF-α in macrophages suggests that Li-ZNPs treatment stimulates and regulates cellular immunity, particularly the activation of pro-inflammatory Th1 cytokines. Moreover, in mice infected with CL, Li-ZNPs significantly reduced the levels of malondialdehyde (MDA), a marker of oxidative stress, and increased the expression levels of the antioxidant enzymes superoxide dismutase (SOD) and glutathione peroxidase (GPx). This suggests that Li-ZNPs have the potential to reduce oxidative damage and enhance antioxidant activity, thereby exerting control over CL in infected mice [93]. Resveratrol (Rsv), curcumin (Cur), and quercetin (QT) (Figure 1 and Figure 3) are natural polyphenols with demonstrated biological activity against several *Leishmania* species, yet their efficacy has been limited [84,104,105]. This limitation is mainly attributed to their low solubility and bioavailability in biological environments [85,106,107]. Nanoemulsions (NEs) are lipid nanocarriers that can efficiently load hydrophobic active compounds. Increased bioavailability and skin retention of some phytochemicals have been observed when encapsulated in NEs [108,109]. Phytosomes (Phys) are phospholipid-based nanotransporters loaded with natural bioactive agents that self-assemble into vesicular structures upon addition of aqueous media. Phys containing polyphenols can increase skin permeability relative to free polyphenols [110]. To improve the efficacy and availability of these polyphenols against *L. major*, resveratrol nanoemulsion (RsvNE), curcumin-loaded AuNPs (Cur-AuNPs), and quercetin-loaded phytosomes (QT-Phys) were developed. RsvNE was less effective after 12 h of treatment than free Rsv. However, in in vivo studies conducted for 21 days in mice treated with 20 mg/kg body weight of RsvNE, an overall reduction in mean lesion size was observed compared to the negative control (PBS), suggesting better efficacy of RsvNE under in vivo conditions (Table 4). Both Rsv and RsvNE inhibit NO production by macrophages, ultimately leading to parasite death [94]. With regard to Cur-AuNPs, they exhibited the capacity to impede the proliferation of promastigotes and amastigotes, albeit with diminished efficacy compared to AmpB (Table 4). In contrast to AmpB, which demonstrated toxicity in macrophages (CC_50_ = 29.80 μg/mL), the evaluated concentrations of Cur-AuNPs did not exhibit significant toxic effects in either cells or mice. Furthermore, additional studies demonstrated that Cur-AuNPs reduced inflammation in the plantar pad of mice infected with CL caused by *L. major* promastigotes following a four-week treatment period, achieving an efficacy comparable to that of AmpB. The leishmanicidal effect of Cur-AuNPs is partially attributed to the induction of the Th1 response, which is characterized by the stimulation of IFN-γ production [95]. In another study, the effectiveness of QT-Phys was evaluated. It was found to kill 100% of *L. major* promastigotes in vitro after 72 h of incubation and completely cure leishmaniasis lesions in vivo after 28 days of treatment. In comparison, free QT was observed to only kill approximately 90% of *L. major* promastigotes (Table 4). The results of the cytotoxicity assays indicated that both QT-Phys and QT exhibited no toxicity at a concentration of 400 μg/mL [96]. It is known that QT inhibits arginase by interacting with Asp129, a crucial amino acid that is part of the active site of the enzyme together with the cofactor (Mn^2+^) [84]. Moreover, studies have revealed that this enzyme is essential for parasite development [111]. Additionally, QT has been demonstrated to activate nuclear factor erythroid 2-related factor 2 (Nrf2), a pivotal regulator of iron homeostasis. This results in a reduction in the availability of free iron, which is essential for the replication and survival of parasites within macrophages [112].

### 2.6. Phytochemical-Loaded Nanosystems to Combat Cutaneous Leishmaniasis Caused by L. mexicana

In México, CL accounts for 99% of cases and is caused mainly by *L. mexicana* [113]. This *Leishmania* species is capable of causing both localized and diffuse cutaneous leishmaniasis. This phenomenon has been associated with the virulence of the parasite, the immune status of the patient, and their genetic background [114]. As mentioned above, Cur and QT exhibit activity against *L. major*. Also, the polyphenols apigenin (Apg) and luteolin (Ltl) (Figure 3) exhibit mild-to-moderate activity against various *Leishmania* species [115,116,117,118]. However, these polyphenols have physicochemical and pharmacokinetic properties that hinder their bioavailability and efficacy. To enhance these characteristics, the compounds were incorporated into nanotransfersomes (NTs), which are bilayer vesicular systems capable of encapsulating amphiphilic, hydrophilic, or hydrophobic drugs in a single nanovesicle. Additionally, NTs are capable of traversing the stratum corneum lipid barrier, thereby facilitating the penetration of drugs into the deeper dermal layers and optimizing their therapeutic activity [119]. Of the four polyphenols—apigenin (Apg), curcumin (Cur), luteolin (Ltl), and quercetin (QT)—loaded in NTs and tested on promastigotes and amastigotes of *L. mexicana*, the apigenin-loaded nanotransfersomes (Apg-NTs) demonstrated the most significant reduction in IC_50_ values (~32%) in both forms of the parasite compared to the free Apg solution, followed by curcumin-loaded nanotransfersomes (Cur-NTs) (Table 5). A formulation of second-generation nanotransfersomes loaded with miltefosine-apigenin (MTF-Apg-SGNTs) was also developed. SGNTs contain at least one phospholipid and two polar lipophilic components, such as an edge activator and a drug-mimicking function. MTF is an amphiphilic compound previously used in lipid nanoparticles for its surfactant properties. Its amphiphilic nature allows it to be positioned close to the phospholipid headgroups in the layers of the nanovesicles, while the polyphenols, which are hydrophobic, are encapsulated within the lipid bilayers [120]. The anti-promastigote activity of MTF-Apg-SGNTs reduced the IC_50_ value up to 6- and 7-fold for Apg-NTs and free Apg, respectively. Higher activity was observed for intracellular amastigotes (Table 5). The anti-promastigote activity of MTF-Apg-SGNTs was markedly enhanced due to the ability of colloidal nanovesicles to traverse the parasite membrane. Furthermore, the efficacy and selectivity of this approach were augmented by the additional benefits of endocytosis in macrophages. In a murine model infected with *L. mexicana*, the complete resolution of the lesion was observed after four weeks of treatment with MTF-Apg-SGNTs gel, in comparison to the application of MTF gel alone. The MTF-Apg-SGNTs gel is capable of penetrating deeply into the dermis layer of infected mice, where it is subsequently engulfed by infected macrophages. This allows for a prolonged release of MTF and Apg, as well as a synergistic interaction between MTF and Apg. Additionally, the Th1-type cytokines (IFN-γ and IL-12) and NO are increased, contributing to the complete clearance of lesions even at a dose that is half that typically required, with no apparent signs of scarring [120].

### 2.7. Phytochemical-Loaded Nanosystems for the Treatment of Cutaneous Leishmaniasis Caused by L. tropica

The use of quercetin-loaded nanotransfersomes in combination with nitazoxanide (NTZ-QT-NTs) was also evaluated for the treatment of CL caused by *L. tropica*, which is characterized by drier and longer-healing skin lesions compared to *L. major* [122]. The results demonstrated that nitazoxanide-loaded nanotransfersomes (NTZ-NTs) and quercetin-loaded nanotransfersomes (QT-NTs) exhibited a markedly reduced IC₅₀ on *L. tropica* promastigotes in comparison to formulations dispersed individually or in combination without NTs. The NTZ-QT-NTs formulation demonstrated a synergistic effect on *L. tropica* promastigotes, as evidenced by an approximately six-fold reduction in IC₅₀ compared to the NTZ-QT dispersion. Additionally, it exhibited superior efficacy compared to NTZ-NTs and QT-NTs. Moreover, the NTZ-QT-NTs gel formulation was tested on infected mice, which exhibited a smaller lesion size compared to the NTZ-QT gel-treated and untreated groups after eight weeks (Table 5). This may be attributed to the capacity of the NTZ-QT-NTs gel to retain the formulation at the site of application, facilitate uptake by macrophages, and effectively release the co-loaded drugs, thereby achieving antileishmanial effects [121].

## 3. Phytochemical Combination Therapy for the Treatment of Trypanosomiasis

Trypanosomiasis is a complex zoonotic disease caused by different species of the protozoan parasite *Trypanosoma* spp., each with distinct clinical manifestations and prognoses. These infections have the potential to affect several different organs. For example, *T. cruzi* has a particular affinity for the heart and digestive system, which can result in the development of American trypanosomiasis, also known as Chagas disease. Conversely, *T. b. gambiense* and *T. b. rhodesiense* primarily affect the central nervous system, resulting in human African trypanosomiasis (HAT), also termed sleeping sickness, in its chronic and acute forms, respectively. Chagas disease is prevalent in Latin America, while sleeping sickness is endemic in Africa, affecting millions of people [123,124,125]. Current treatments for HAT include nifurtimox-eflornithine combination therapy (NECT) and fexinidazole (FX). However, the application of these treatments is mainly limited to the gambiense form of HAT, while *T. b. rhodesiense* is treated with melarsoprol (Mel B), a drug that can induce reactive encephalopathy. The first-line drugs for Chagas disease are benznidazole (BNZ) and nifurtimox (NF), which have significant limitations such as low curative efficacy, especially in the chronic phase, and toxicity (Table 1) [21,24,126]. It is therefore imperative to develop new drugs that are less toxic and more effective at all stages of the disease. Consequently, the inhibition of rhodesain, a cysteine protease of *T. b. rhodesiense*, represents a key target in the development of new therapies for HAT. This enzyme facilitates the entry of the parasite into the brain by crossing the blood–brain barrier, thereby triggering the neurological phase of the disease. Furthermore, rhodesain plays a crucial role in the parasite’s evasion of the host immune system, involving surface glycoprotein turnover and immunoglobulin degradation. Additionally, the enzyme displays proteolytic activity within lysosomes, facilitating the degradation of intracellular proteins from both the host and the parasite [127,128]. In this context, several potential inhibitors of rhodesain have been investigated. The dipeptide nitrile CD24 is a competitive inhibitor that binds directly to the active site of the enzyme [129]. Conversely, the vinyl ketone RK-52 functions as a highly potent irreversible inhibitor of rhodesain [130], while Cur operates as a non-competitive inhibitor of rhodesain [131]. To enhance the efficacy of these synthetic compounds, they were combined with curcumin (CD24-Cur and RK-52-Cur). The results demonstrated that CD24-Cur exhibited superior antitrypanosomal activity compared to CD24 alone. In contrast, the combination of RK-52 and Cur did not result in a significant enhancement in activity compared to the individual compounds (Table 6). However, it was observed that both CD24-Cur and RK-52-Cur demonstrated remarkable synergistic action against *T. b. rhodesiense* rhodesain [132,133].

With regard to Chagas disease, the Rsv molecule and the resveratrol–benznidazole (Rsv-BNZ) combination, administered at 100 mg/kg, demonstrated an absence of efficacy in parasite reduction, with a lower efficacy than that observed with BNZ (100 mg/kg). Nevertheless, Rsv demonstrated a more pronounced effect than the Rsv-BNZ combination, reducing ROS production and positively regulating purinergic receptors. This mechanism may serve as a compensatory mechanism for parasite clearance and the reduction of oxidative damage in the cerebral cortex of infected mice [135].

### Phytochemical-Loaded Nanosystems for the Treatment of Chagas Disease

Curcumin (Cur) has demonstrated useful properties in the control of *T. cruzi*. In *T. cruzi*-infected mice, it reduced parasitemia and parasitism of infected heart tissue and reduced cardiovasculopathy [136,137]. Based on considerations of pharmacokinetics and/or physicochemical properties of Cur, researchers have chosen to use nanotechnology. Oral administration of curcumin-benznidazole-loaded nanoparticles (Cur-BNZ-NPs) at suboptimal doses reduced myocardial parasite load, cardiac hypertrophy, inflammation, and fibrosis in mice with long-term *T. cruzi* infection. Cur-BNZ-NPs were also shown to negatively regulate the expression of myocardial proinflammatory cytokines and chemokines and to reduce the levels and activity of matrix metalloproteinases (MMP-2, MMP-9) and inducible enzymes such as iNOS, which are involved in leukocyte recruitment and cardiac remodeling [138].The formulation of Theracurmin, composed of nanoparticles containing 10% w/w Cur, 2% other natural curcuminoids (such as demethoxycurcumin and bisdemethoxycurcumin), 46% glycerin, 4% ghatti gum, and 38% water, was evaluated. The mean particle size of Theracurmin is 0.19 µm, compared to 22.75 µm for conventional Cur. Additionally, Theracurmin reaches peak plasma concentration within 1.5–3 h, whereas Cur powder requires 18.4 to 20.5 h in humans. Theracurmin exhibits a 27-fold higher absorption rate in humans than Cur [139,140]. In mice experimentally infected with *T. cruzi*, Theracurmin treatment resulted in a reduction in the parasitemia curve, without affecting the survival rate of the animals. Furthermore, the treatment protected against loss of body mass, although to a lesser extent than BNZ. Infection elevated plasma creatine kinase (CK) concentrations in the animals, indicating muscle damage caused by the parasite. Theracurmin administration reduced CK levels during the acute phase of *T. cruzi* infection, showing a pattern similar to BNZ therapy. The presence of *T. cruzi* in cardiac and skeletal muscle tissue is known to increase inflammatory mediators such as TNF, IL-6, IL-15, and chemokine (C-C motif) ligand 2 (CCL2). However, Theracurmin treatment reduces CCL2 and IL-15 production, suggesting a decrease in mononuclear cell recruitment to the inflammatory focus during the later stages of infection [141]. CCL2, a key chemokine in the parasite-associated inflammatory response at various stages of infection, and IL-15, involved in T and NK cell proliferation and implicated in tissue damage in chronic chagasic cardiomyopathy, are both negatively regulated by Theracurmin [142]. Another study highlights that in a murine model of *T. cruzi* infection, Theracurmin exerts protective effects by regulating IL-15 synthesis, although it does not affect CCL2 levels in damaged testicular tissue [143]. The mechanism underlying these protective effects of Theracurmin relates to its ability to negatively regulate MAPK activity and nuclear factor kappa B (NF-κB) signaling [144,145]. These findings suggest that Theracurmin may offer an effective therapeutic intervention by attenuating inflammation and tissue damage associated with chronic *T. cruzi* infection. Lychnopholide (LYC) (Figure 3) is a lipophilic sesquiterpene lactone that has been isolated from the plant *Lychnophora trichocarpha* (Spreng.) of the Asteraceae family. LYC is a molecule with low water solubility and high lipophilicity (log P = 5.03) and, in its free form, has a short half-life in mice of 21 min when administered intravenously [146]. To enhance its bioavailability and efficacy, it was encapsulated in biodegradable poly (D, L-lactide) (PLA)-block-polyethylene glycol (PEG) nanocapsules (NC). The LYC-PLA-PEG NC (LYC-NC) formulation was evaluated in vitro on *T. cruzi* epimastigotes and intracellular amastigotes, yielding comparable outcomes (IC_50_ = 1.62; IC_50_ = 0.05 µM) to free LYC and marginally superior to those of BNZ. The SI value was found to be approximately 30% higher for LYC-NC in comparison to free LYC [147]. In mice infected with a BNZ- and NF-resistant strain of *T. cruzi*, the efficacy of treatment was evaluated in both the acute (AF) and chronic (CF) phases of infection. The results obtained according to the classical cure criteria indicate that treatment with LYC-NC at 12 mg/kg/day resulted in a 75% (AF) and 88% (CF) cure rate, whereas no cure was observed in animals treated with free LYC (12 mg/kg/day), free BNZ (100 mg/kg/day), or in controls [148]. The efficacy of the LYC-NC formulation is partly due to the modification of the NC surface by PEG. The PEG chains delay the rapid clearance of NC from the bloodstream by macrophages, thereby prolonging the plasma half-life associated with the drug [149,150]. Consequently, these nanoparticles have the potential to extravasate into *T. cruzi*-infected tissues, particularly at inflammatory sites [151]. The localization of NC in the cytoplasm of mammalian cardiac cells, where amastigotes multiply, and in *T. cruzi* trypomastigote, suggests that LYC-NC can effectively deliver LYC to desired targets [147].

## 4. Combination Therapy with Phytochemicals for the Treatment of Schistosomiasis

Schistosomiasis, a globally prevalent helminthiasis, is predominantly caused by *S. mansoni*, *S. japonicum*, or *S. haematobium* [152]. The current standard treatment, praziquantel (PZQ), exhibits efficacy against adult worms of the parasite but not against eggs and developing larvae (schistosomula), resulting in suboptimal cure rates and a propensity for rapid reinfection [28]. Repeated therapy with PZQ is necessary for effective disease control. However, the emergence of drug resistance [153] raises concerns, reinforcing the need for alternative chemotherapeutic strategies, such as combinations of pharmacological agents and the implementation of nanodrugs, to optimize clinical management and reduce the impact of the disease. In this context, a recent study evaluated the antischistosomal activity of Rsv and its capacity to enhance the efficacy of the anthelmintic drugs PZQ and artesunate (As) against newly transformed schistosomula (NTS) of *S. mansoni*. The results demonstrated that Rsv exhibited modest antischistosomal activity against NTS. This effect may be attributed to its capacity to enhance oxidative stress, as previously documented [154]. It is noteworthy that the combination of As or PZQ with Rsv markedly enhanced the antischistosomal effect compared to a single treatment, as illustrated in Table 6. The resveratrol–praziquantel (Rsv-PZQ) and resveratrol–artesunate (Rsv-As) combinations demonstrated a synergistic effect, indicating an increased action on anthelmintic drug targets or concomitant action on different targets [134]. Ultrastructural analysis revealed that the NTS treated with As-Rsv experienced extensive damage, including disruption of the tegument, extensive lysis of subtegumental regions with the formation of numerous vacuoles of various sizes, and loss of the basement membrane [134]. The schistosome tegument plays a pivotal role in the parasite’s defense against the host’s immune response and the absorption of essential nutrients [155]. Consequently, the disruption induced by Rsv-As could harm the parasite’s capacity to maintain its nutrition and evade the host’s immune response.

### Phytochemical-Loaded Nanosystems to Treat Schistosomiasis

In the field of antiparasitic research, several natural compounds have demonstrated efficacy against *S. mansoni*, including carvacrol (Cv), licochalcone A (Lico A), curcumin (Cur), and (±)-licarin-A (Lic-A) (Figure 1 and Figure 3). Each of these compounds possesses distinctive properties and presents a unique set of challenges in their therapeutic application. Cv, a natural monoterpene with antimicrobial properties, exhibits slow intestinal absorption and high gastrointestinal retention, which limits its effectiveness against schistosomes [156,157]. To enhance its efficacy, an oral carvacrol nanoemulsion (CvNE) was formulated and administered to mice infected with immature (prepatent infection) or adult (patent infection) *S. mansoni*. A single dose of CvNE (200 mg/kg) resulted in a significant reduction in worm burden and egg production in both stages of infection when compared to free Cv. This highlights the improved efficacy of the nanoformulation (Table 7) [158]. In prepatent infection, CvNE demonstrated superior efficacy to PZQ, which has limited activity against the juvenile stage [49,158]. The precise mechanism of action of Cv against schistosomes remains uncertain. However, its high hydrophobicity and chemical characteristics facilitate cell penetration and interaction with intracellular proteins [147]. Additionally, its acaricidal activity has been linked to enhanced antioxidant enzyme activity. Furthermore, there is evidence that CvNE inhibits acetylcholinesterase and reduces levels of glutathione and malondialdehyde [159,160]. Similarly, Lico A, a natural chalcone, exhibits potent biological properties against parasites, including *S. mansoni* [161,162,163]. However, its low water solubility and hemolytic effects in erythrocytes limit its therapeutic application [164,165]. SLNs use improves drug protection, physical stability, and the ability to control and direct the administration of the drug [166]. To overcome these limitations, licochalcone A was encapsulated in solid lipid nanoparticles (Lico A-SLNs) and evaluated both in vitro and in vivo. Both free Lico A and Lico A-SLNs were found to have schistosomicidal activity against adult worms. Free Lico A achieved 100% mortality within 24 h, while Lico A-SLNs required 48 h due to its controlled release. Free Lico A showed cytotoxicity (CC_50_ = 20 μM) and hemolytic effects (RBC_50_ < 1 mM), which were significantly mitigated by Lico A-SLNs (CC_50_: 956.2 μM; RBC50 > 5 mM). In mice, intraperitoneal administration of free Lico A and Lico A-SLNs similarly reduced worm and immature egg burdens; however, a greater effect was observed on mature eggs (Table 7). The observed effect may be because both free Lico A and Lico A-SLNs damage the tegument of *S. mansoni* male worms, which is crucial for their survival and immune evasion [155,166].

On the other hand, Cur is active against several parasites, including *S. mansoni* [105,137,169,170]. Therefore, Cur-AuNPs were tested in mice infected with *S. mansoni* cercariae. Monotherapy of schistosomiasis with Cur was less effective than with PZQ. In contrast, the curcumin–praziquantel (Cur-PZQ) combination achieved a significant reduction in worm burden, accompanied by a marked reduction in intestinal and liver egg counts. Treatment with Cur-AuNPs did not show a significant reduction in worm burden, but a more significant reduction was achieved when combined with PZQ. The addition of PZQ also significantly increased the reduction in intestinal and liver eggs compared to treatment with Cur-AuNPs alone [160]. These effects may be related to the observed transcriptional repression of the Notch and TGF-β pathways by Cur in *S. mansoni* [171]. Lic-A is a naturally occurring dihydrobenzofuranic neolignan with reduced solubility that shows activity against parasites responsible for Chagas disease, leishmaniasis, and schistosomiasis [172,173]. Therefore, pure (±)-licarin-A neolignan was incorporated into poly-Ɛ-caprolactone nanoparticles (Lic-A-PCLs) and tested in mice infected with cercariae of *S. mansoni*. Lic-A-PCLs reduced the blood worm burden similarly to free Lic-A but to a lesser extent than PZQ. However, treatment with free Lic-A reduced the number of eggs in the liver and spleen to a greater extent than Lic-A-PCLs and PZQ (Table 7) [161]. Although the mechanism of action of Lic-A as a schistosomicide is unclear, its activity against other parasites is known to be associated with immunomodulatory activity (IL-6 and IL-10) and ultrastructural changes [174,175]. However, further research is needed to fully understand how Lic-A exerts its schistosomicidal effect in vivo.

## 5. Future Trends

Research on combination therapies and phytochemical-loaded nanosystems for the treatment of parasitic diseases such as leishmaniasis, trypanosomiasis, and schistosomiasis is opening new avenues for more effective and safer treatments. In this context, several key trends are identified that are likely to dominate the field in the coming years: (a) Development of multifunctional nanoparticle-mediated combination therapies. Future efforts will be directed towards the creation of multifunctional nanosystems that not only improve the solubility and bioavailability of combined phytochemicals but also allow for controlled and targeted release. These nanosystems could be designed to release drugs in response to specific stimuli from the parasite microenvironment, such as changes in pH or the presence of parasitic enzymes, which would maximize therapeutic efficacy. In addition, these formulations are expected to include features to overcome biological barriers and enhance selective accumulation in infected tissues, which is crucial for the treatment of late-stage parasitic infections [176]; (b) co-delivery of chemotherapeutics and immunostimulants using the carrier-free strategy called drug-delivering-drug. This strategy improves co-administration by using drug crystals of insoluble drugs as a carrier to deliver the second drug, such as biopharmaceuticals and low-molecular-weight compounds. This approach could increase the drug load up to 20 times more than conventional drug carriers. This could be a valuable strategy to improve sustained drug release over time after internalization and enhance the host immune response in treating parasitic diseases [177]; and (c) transition to clinical trials and validation in humans. Although preclinical research has shown promising results, the next critical phase will be the validation of these combination therapies and nanoformulations in human clinical trials. As nanoparticles (NPs) may impose risks to patients due to their ability to pass some biological barriers, they may exert potentially fatal toxic effects, especially in essential organs, including the brain, liver, and kidneys. Therefore, it is necessary to evaluate not only safety and efficacy in different populations but also to investigate optimal dosing, possible accumulation in non-target sites, elimination, excretion from the body, and other factors that can potentially lead to life-threatening complications [178]. Future clinical trials are also expected to focus on the treatment of both acute and chronic phases of parasitic diseases, addressing the urgent need for new therapies for resistant or difficult-to-treat diseases.

## 6. Conclusions

Combination therapy with phytochemicals and the use of nanosystems loaded with these compounds represent a significant advance in the treatment of various parasitic diseases such as leishmaniasis, trypanosomiasis, and schistosomiasis. For VL, compounds such as Lup and UA have been shown to improve therapeutic efficacy, reduce parasite burden, and modulate the immune response when combined with conventional drugs such as AmpB and MTF. Encapsulation of these compounds in NLCs further enhances their solubility and efficacy, overcoming the limitations of conventional treatments and minimizing toxicity. Likewise, in CL, the combination of UA-Glu and the use of BA-KNPs and MTF-APG co-loaded SGNTs not only significantly reduces the size of skin lesions but also enhances the immune response without causing significant toxicity. In the treatment of HAT caused by *T. b. rhodesiense*, where there are few effective pharmaceutical agents, the combination of CD24-Cur and RK52-Cur has been shown to have a synergistic effect against *T. b. rhodesiense* rhodesain. These options could be further explored in the development of drugs for the neurological phase of the disease. On the other hand, the use of Cur-BNZ NPs may be beneficial in the chronic phase of Chagas disease, as it has been shown to reduce myocardial parasite burden, cardiac hypertrophy, inflammatory response, and fibrosis in mouse models. Meanwhile, in the acute phase of the disease, Theracurmin could be an interesting option as it lacks the toxic effects of BNZ. In the context of schistosomiasis, where there is an urgent need for treatments that cover juvenile forms of *S. mansoni* and prevent reinfection, the combination of Rsv-As and Rsv-PZQ, as well as CvNE, represent promising therapeutic options due to their effective activity against *S. mansoni* schistosomula. The combination of phytochemicals and nanosystems represents a novel and effective strategy for the treatment of parasitic diseases, including leishmaniasis, trypanosomiasis, and schistosomiasis. The enhanced solubility, efficacy, and selectivity of these treatments, coupled with their reduced toxicity, represent a substantial advancement over conventional treatments. Future research should prioritize the optimization of these combinations and the conduction of clinical studies to validate the efficacy of these therapies in humans. This will facilitate the development of safer and more effective alternatives for the control of these NTDs.

## Figures and Tables

**Figure 1 pharmaceutics-16-01239-f001:**
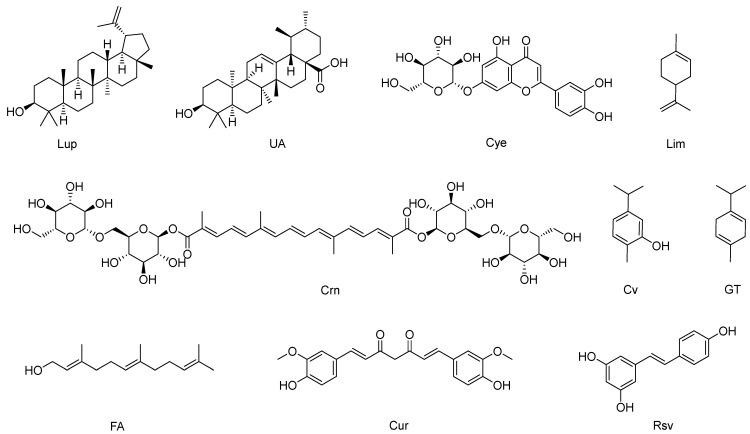
Phytochemicals used in combination therapy for the treatment of neglected tropical diseases. Lup: Lupeol; UA: ursolic acid; Cye: cynaroside; Lim: limonene; Crn: crocin; Cv: carvacrol; GT: gamma-terpinene; FA: farnesol; Cur: curcumin; Rsv: resveratrol.

**Figure 2 pharmaceutics-16-01239-f002:**
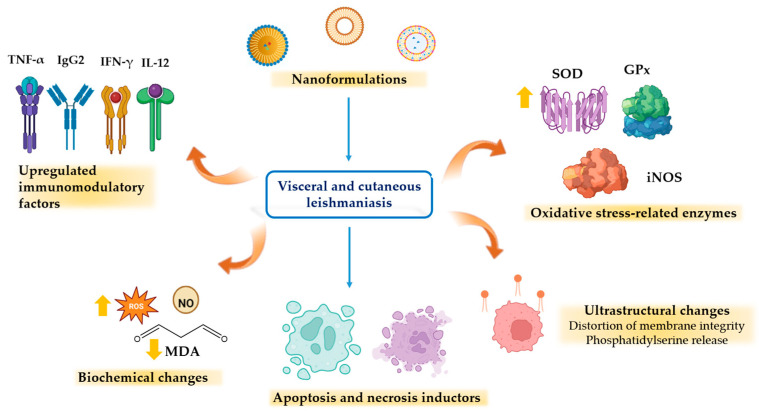
Main molecular mechanisms of phytochemical-loaded nanoformulations against *Leishmania* spp.

**Figure 3 pharmaceutics-16-01239-f003:**
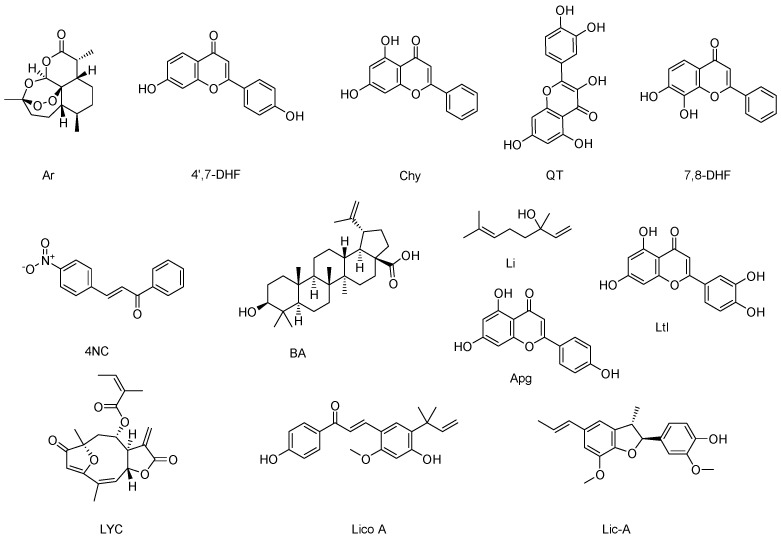
Phytochemical-loaded nanosystems for the treatment of neglected tropical diseases. Ar: artemisinin; 4′,7-DHF: 4′,7-dihydroxyflavone; Chy: 5,7-dihydroxyflavone; QT: quercetin; 7,8-DHF: 7,8-dihydroxyflavone; 4NC: 4-Nitrochalcone; BA: betulinic acid; Apg: apigenin; Ltl: luteolin; LYC: Lychnopholide; Lico A: licochalcone A; Lic-A: (±)-licarin-A.

**Table 1 pharmaceutics-16-01239-t001:** Principal pharmaceutical agents employed in the treatment of neglected tropical diseases.

Drug	Mechanism of Action	Advantages	Disadvantages	Ref.
AmpB (Leishmanicidal)	The drug interacts with ergosterol, the main membrane sterol of the parasite, forming pores in the plasma membrane, changing the permeability to ions and metabolites, and generating reactive oxygen species (ROS).	Amp B has an excellent cure rate (~100%) at a dose of 0.75–1 mg/kg for 15–20 intravenous infusions daily or every other day for visceral leishmaniasis (VL).	It causes severe nephrotoxicity and hematological toxicity, hemolysis, liver damage, nausea, and fever. Resistance has been found in several *Leishmania* strains and clinical isolates.	[10,11,12,13,14]
MTF (Leishmanicidal)	Interferes with membrane lipids and mitochondrial function of the parasite.	MTF is the only oral drug approved for the treatment of both VL and cutaneous leishmaniasis (CL). 77% efficiency in CL.	Causes gastrointestinal intolerance and teratogenicity in pregnant women. Resistance in clinical isolates. Unaffordable.	[13,15,16,17,18]
PNT (Trypanocidal)	The drug accumulates in very high concentrations in the mitochondria of the trypanosome, where it binds to DNA in the kinetoplast, inhibiting both replication and transcription.	PNT has therapeutic effects in early-stage human African trypanosomiasis (HAT).	PNT is only used to treat early-stage trypanosomiasis caused by *Trypanosoma brucei gambiense.* PNT resistance.	[19,20]
Mel B (Trypanocidal)	Mel B is rapidly metabolized to melarsen oxide. Melarsen oxide can cause loss of parasite motility. Mel B can form a Mel T complex. Mel T is a competitive inhibitor of the antioxidant enzyme trypanothione reductase.	Mel B displays no toxic effect on the optic nerve. Mel B is the only drug effective in stage II of *T. b. rhodesiense* sleeping sickness.	Causes reactive encephalopathy that can be fatal in up to 10% of treated patients. Resistance due to genetic modifications in the aquaglyceroporin 2 (AQP2) gene.	[21,22,23]
NECT (Trypanocidal)	Eflornithine: irreversible ornithine decarboxylase inhibitor.	For the treatment of stage II HAT by *T. b. gambiense*: the cure rate for NECT is ~96.5%, compared to 9.6% for eflornithine. NECT requires a much simpler and less intensive treatment regimen than eflornithine monotherapy.	NECT is ineffective against *T. b. rhodesiense* and requires 14 days of systemic treatment with eflornithine.	[24,25,26,27]
FX (Trypanocidal)	It produces a pronounced defect in DNA synthesis that reduces the parasite population in the S phase of the cell cycle.	Oral administration. Effective (99%) in the treatment of stage I and early stage II HAT (*T. b. gambiense*).	FX is only used to treat trypanosomiasis caused by *T. b. gambiense*. Moderate adverse events.	[25,26,28]
BNZ (Trypanocidal)	BNZ is activated by trypanosomal nitroreductase I and produces glyoxal dialdehyde, which blocks normal parasite DNA/RNA function, increasing vulnerability to life-cycle oxidative damage.	Pretreatment with BNZ is associated with less advanced heart disease, lower prevalence of detectable parasitemia, and lower mortality in patients with chronic Chagas disease.	Not effective in chronic cases; has side effects and has been associated with resistance.	[29,30]
NF (Trypanocidal)	NF, activated by nitroreductases, induces nitro anion radicals. These react with oxygen to produce free radicals that damage *T. cruzi*. They also inhibit DNA synthesis and promote DNA degradation.	NF is moderately effective in the acute phase of Chagas disease.	NF is not effective in chronic cases. Its high toxicity is associated with intrauterine developmental delay and reduced body weight in rat and mouse fetuses. In addition, it induces chromosomal aberrations in infected children.	[31]
PZQ (Schistosomicidal)	PZQ causes altered calcium homeostasis, which in turn induces muscle contraction and alterations in the integument.	Effective against all *Schistosoma* species infecting humans, with cure rates often exceeding 80%. PZQ has some mild and transient side effects.	Ineffective against young forms of *Schistosoma mansoni.* Limited effect on granulomatous lesions. Significantly reduced susceptibility to PZQ in foci of endemicity.	[32,33]

AmpB: amphotericin B; MTF: miltefosine PNT: pentamidine; Mel B: melarsoprol; NECT: nifurtimox-eflornithine; FX: fexinidazole; BNZ: benznidazole; NF: nifurtimox; PZQ: praziquantel.

**Table 2 pharmaceutics-16-01239-t002:** Combination therapy with phytochemicals for the treatment of visceral and cutaneous leishmaniasis.

Combination Therapy/Monotherapy (mg/kg)	Model/Parasite	Parasite Load (%)/IC₅₀ (µM)/Footpad Thickness	CC_50_ (µM)	Major Outcome	Ref.
Lup (3) + AmpB(0.1)	BALB/c mice were infected with stationary-phase *L. donovani* promastigotes (i.v., 1 × 10^7^/mouse).	Liver:89.22; spleen:83.58		A suboptimal dose combination of Lup and AmpB significantly reduced hepatic and splenic parasite load, increased Th1 cytokines, and suppressed Th2 cytokines.	[51]
Lup (3)		Liver:6.85; spleen:4.38			
AmpB (0.1)		Liver:8.9; spleen:7.06			
UA (1.0) + AmpB (1.0)	Golden hamsters were inoculated with *L. infantum* (MHOM/BR/72/46) promastigotes (i.p., 2 × 10^7^/hamster).	Liver:86.5; spleen:60.7		Animals treated with UA (1.0 mg/kg) significantly reduced splenic and hepatic parasitism compared to AmpB (0.2 mg/kg) and combination therapy.	[52]
UA (1)		Liver:99.9; spleen:99.8			
AmpB (0.2)		0%			
Cye (20 µM) +MTF (4 µM)	*L. donovani* (MHOM/IN/83/AG83) promastigotes in the log phase were incubated (5 × 10^6^ parasites).	> 50		Cye demonstrated a better response when used in combination with low concentrations of MTF than with monotherapy using either Cye or MTF.	[53]
Cye		49.49 ± 3.51			
MTF		6.43 ± 0.50			
Cye (20 µM) +MTF (4 µM)	THP-1-differentiated macrophages were infected with *L. donovani* (1 macrophage:10 parasites)	>80			
Cye (20 µM)		50	65.33 ± 5.27		
MTF (4 µM)		80	20.39 ± 1.69		
Crn+ AmpB	*L. major* (MRHO/IR/75/ER) promastigotes were incubated (1 × 10^6^ parasites).	229.6 ± 6.2		The combination Crn + AmpB had a greater effect on inhibiting intra-macrophagic promastigote multiplication than monotherapy with Crn, by inhibiting L-ARG levels, potentiating immune response, and arresting cell cycle growth.	[54]
Crn		382.7 ± 23			
AmpB		293.5 ± 71			
Crn+ AmpB	J774 murine macrophages were infected with *L. major* promastigotes (1 macrophage:10 parasites).	24.5 ± 6.4 (SI = 18.1)	328.7		
Crn		95.8 ± 18 (SI = 20.2)	1935.3		
AmpB		43.7 ± 13 (SI = 7.5)	328.7		
UA (1.0) + Glu (2.0)	BALB/c mice were infected with *L. amazonensis* promastigotes in the stationary phase of growth (2 × 10^7^ parasites/mL) in the tail.	97.6		Treatment with Glu (2.0 mg/kg) alone showed no reduction in skin parasitism in infected BALB/c mice. However, when Glu (2.0 mg/kg) was combined with UA (1.0 mg/kg), a 97.6% decrease in skin parasitism was observed. A positive increase in IFN-γ production and a decrease in IL-4 production were also observed.	[52]
UA (1.0)		96.2			
Glu (2.0)		0			
Lim + Car 1:4	Macrophages were incubated with *L. major* promastigotes in stationary phase (1 macrophage:10 parasites).	~20.59 (SI = 19.0)	~391.55	Lim + Car 1:4 demonstrated no remarkable activity against L. major amastigotes and promastigotes.	[55]
Lim		~194.58 (SI = 5.9)	~1166.55		
Car		~64.83 (SI = 9.6)	~626.68		
AmpB		~0.42 (SI = 22.4)	~9.46		
Lim + Car 4:1	*L. major* (MHOM/IL/80/Friendlin) promastigotes in the log phase were incubated (1 × 10^6^ parasites).	~47.12 (SI = 8.30)			
Lim		~117.43 (SI = 9.93)			
Car		~38.61 (SI = 16.23)			
AmpB		~0.67 (SI = 14.11)			
GT (10,000 µM) + Glu (750µM)	Macrophages were incubated with *L. major* promastigotes in stationary phase (1 macrophage:10 parasites).	>50%		GT + Glu combination exhibited antagonistic interaction against *L. major* amastigotes.	[56]
GT		25,890 (SI = 3.17)	82,100		
Glu		1280 (SI = 8.08)	10,350		
FA (250 mM) + Par-cream	BALB/c mice were injected with stationary-phase *L. major* (MHOM/SU/73/5ASKH) promastigotes (i.d., 10^6^–10^7^) in each hindfoot.	ND		The combined therapy of F (250 mM) + Par-cream in topical administration showed an antagonistic effect, with a greater plantar pad thickness observed than in the monotherapy application.	[57]
FA (250 mM)		2.16 mm (footpad thickness)			
P-cream		~2.4 mm (footpad thickness)			

Lup: lupeol; AmpB: amphotericin B; UA: ursolic acid; Cye: cynaroside; MTF: miltefosine; Crn: crocin; Glu: glucantime; Lim: limonene: Car: carvacrol; GT: gamma-terpinene; FA: farnesol; Par-cream: paromomycin in cream; SI: selectivity index; CC_50_: cytotoxic concentration 50%; i.p.: intraperitoneal; i.d.: intradermally.

**Table 3 pharmaceutics-16-01239-t003:** Phytochemical-loaded nanosystems for the treatment of visceral leishmaniasis.

Nanosystem/Monotherapy (mg/kg)	Model/Parasite	Parasite Load (%)/IC_50_ (µg/mL)	Physicochemical Properties/Cytotoxicity	Major Outcome	Ref.
Lup-NCLs (5)	Golden hamsters were infected with *L. infantum* (MHOM/BR/72/46) promastigotes in the stationary phase of growth (i.p., 2 × 10^7^).	Liver:99.9; spleen:99.9	PS (nm) = 265.3 ± 4.6; PDI = 0.21 ± 0.011; ZP (mV) = −37.2 ± 0.84 ; EEn (%) = 84.04 ± 0.57	Treatment with Lup-NLCs in infected hamsters significantly reduced the number of parasites in the spleen and liver, with effects comparable to those of AmpB, and also improved organ preservation and elevated anti-*Leishmania* IgG2 levels.	[61,63]
Lup (5)		Liver:90.2; spleen:94.4	Log Sw = −8.57; Log P = 7.6; PS (nm) = 266.3 ± 2.6; PDI = 0.16 ± 0.013; ZP (mV) = −26.5 ± 1.18		
AmpB (5)		Liver: 99.7; spleen:99.4			
UA-NLCs (1.25)	Golden hamsters were infected with *L. infantum* (MHOM/BR/72/46) promastigotes (i.p., 2 × 10^7^).	Liver:99.78; spleen:98.63	PS (nm) = 266.3 ± 5.4; PDI = 0.18 ± 0.022; ZP (mV) = 29.26 ± 1.16; EEn (%) = 59.71 ± 0.2	Administration of UA-NLC to infected hamsters significantly reduced the parasite load in the spleen and liver, outperforming free UA and AmpB, and enhanced the immune response without the renal toxicity observed with AmpB.	[64]
UA (1.25)		Liver:90.04; spleen:74.20	Log Sw = −7.23; Log P = 5.93		
AmpB (5)		Liver:~99; spleen:~98	AmpB increased AST, creatinine and urea concentrations		
Ar-SLNs (20)	BALB/c mice were inoculated with *L. infantum* (MCAN/IR/07/Moheb-gh) promastigotes in the stationary phase (i.v., 2.5 × 10^7^).	Liver:84.7 ± 4. 9; spleen: 85.0 ± 3.1	PS (nm) = 222.0 ± 14.0; PDI = 0.5; EE (%) = 75	Ar-SLNs reduce parasite load by ~30% in a murine model of visceral leishmaniasis, improving over free Ar.	[65]
Ar (20)		Liver:~60; spleen:~60			
4′,7-DHF-MWCNT	*L. donovani* (MHOM/IN/80/DD8) promastigotes in the log phase were incubated (1 × 10^6^).	0.05 ± 0.01	ZP (mV) = 35; LE (%) = 84.28 ± 0.55	The 4′,7-DHF-MWCNT formulation showed ~28-, 11-, and 38-fold higher activity on promastigotes, intracellular amastigotes, and amastigotes compared to administration of 4′,7-DHF alone.	[66,67]
4′,7-DHF		1.42	Log Po/w = 2.40 ; LogS = −4.03		
MWCNT		0.93 ± 0.17			
4′,7-DHF-MWCNT	Macrophages were incubated with *L. donovani* promastigotes at a 1:10 ratio.	0.08 ± 0.02 (SI = ~95)	CC₅₀ = 7.62 ± 0.70 μg/mL		
4′,7-DHF		0.87 (SI = > 1000)	CC₅₀ = ~1131 μg/mL		
MWCNT		5.15 ± 0.95 (SI = ~3)	CC₅₀ = 11.7 ± 1.7 μg/mL		
4′,7-DHF-MWCNT	*L. donovani* amastigotes in the log phase were incubated (1 × 10^6^).	0.07 ± 0.01			
4′,7-DHF		2.78			
MWCNT		3.01 ± 0.23			
4′,7-DHF-AuNPs	L. donovani (MHOM/IN/80/DD8) promastigotes in the log phase were incubated (1 × 10^6^).	0.12 ± 0.02	PS (nm) = 5.8 ± 0.1; PDI = 0.412; ZP (mV) = 60; EEn (%) = 80.79 ± 2.17	The 4′,7-DHF-AuNPs formulation demonstrated ~12, 7, and 25 times more activity on promastigotes, intracellular amastigotes, and amastigotes than free 4′,7-DHF.	[67,68]
	Macrophages were incubated with *L. donovani* promastigotes at a 1:10 ratio.	0.12 ± 0.36 (SI = 24)	CC₅₀ = 2.95 ± 0.46 µg/mL		
	L. donovani amastigotes in the log phase were incubated (1 × 10^6^).	0.11 ± 0.02			
4′,7-DHF-AgNPs	*L. donovani* (MHOM/IN/80/DD8) promastigotes in the log phase were incubated (1 × 10^6^).	0.84 ± 0.14	PS (nm) = 10; PDI = 0.62; ZP (mV) = 40; EEn (%) = 64.13 ± 0.83	The 4′,7-DHF-AgNPs formulation demonstrated ~2, 4, and 11 times more activity on promastigotes, intracellular amastigotes, and amastigotes than free 4′,7-DHF.	
	Macrophages were incubated with *L. donovani* promastigotes at a 1:10 ratio.	0.21 ± 0.85 (SI = 23)	CC₅₀ = 4.95 ± 0.69 µg/ml		
	*L. donovani* amastigotes in the log phase were incubated (1 × 10^6^).	0.26 ± 0.58			
Chy-AuNPs	J774A.1 macrophages were infected with *L. donovani* (MHOM/IN/80/DD8) promastigotes at a 1:10 ratio.	0.8 ± 0.08	PS (nm) = 20 ± 0.14; LE (%) = 90.86 ± 0.86	The Chy-AuNPs formulation exhibited ~3 times more activity on intracellular amastigotes than free Chy.	[69]
Chy		2.19 ± 0.41			
QTNE	*L. donovani* (clinical isolate) promastigotes were incubated (1 × 10^6^).	1.99 (48 h)	PS (nm) = 38.90 ± 3.33; PDI = 0.290 ± 0.010	QTNE significantly reduced the number of treated promastigotes by directly enhancing ROS production, leading to death by late apoptosis/necrosis.	[70,71]
QT		~1.99 (48 h)			
7,8-DHF-AuNPs	RAW264.7 macrophages were infected with *L. donovani* (MHOM/IN/1983/AG83) promastigotes.	~6.35	PS (nm) = 35 ± 7.4; PDI = 0.19 ± 0.01; ZP (mV) = 34.1; EE (%) = ~8–10	The antileishmanial efficacy of 7,8-DHF-AuNPs is much higher than that of free 7,8-DHF. The 7,8-DHF-AuNPs were equally effective against sensitive and resistant strains of *L. donovani*. Treatment with 7,8-DHF-AuNPs inhibits arginase.	[72,73]
7,8-DHF		~14.49	Water solubility (μg/mL) = 7.12; bioaccessibility (%) = 18.06		
7,8-DHF-AuNPs	*L. donovani* promastigotes were incubated in log phase (1 × 10^6^)	~11.69			
7,8-DHF		~35.59			
7,8-DHF-AuNPs	AmpB-resistant clinical isolate of *L. donovani*.	~10.16			
7,8-DHF		~38.13			
7,8-DHF-AuNPs	SAG-resistant clinical isolate of *L. donovani.*	~8.89			
7,8-DHF		~40.67			

Lup: lupeol; AmpB: amphotericin B; UA: ursolic acid; NLCs: nanostructured lipid carriers; solid lipid nanoparticles (SLNs); artemisinin: Ar; MWCNT: multi-walled carbon nanotubes; 4′,7-DHF: 4′,7-dihydroxyflavone; AuNPs: gold nanoparticles; AgNPs: silver nanoparticles; Chy: 5,7-dihydroxyflavone; NE: nanoemulsion; QT: quercetin; 7,8-DHF: 7,8-dihydroxyflavone; SAG: sodium antimony gluconate; SI: selectivity index; CC_50_: cytotoxic concentration 50%; AST: aspartate aminotransferase; PS: particle size; PDI: polydispersity Index; ZP: zeta potential; EEn: encapsulation efficiency: EE: entrapment efficiency; LE: load efficiency; LogS: logarithm of the solubility in water; Log Po/w: logarithm of the partition coefficient; i.p.: intraperitoneal; i.v.: intravenous.

**Table 4 pharmaceutics-16-01239-t004:** Phytochemical-loaded nanosystems for treatment of cutaneous leishmaniasis caused by *L. amazonensis* and *L. mayor*.

Nanosystem/Monotherapy (mg/kg)	Model/Parasite	Dead (%)/IC_50_ (µg/mL)	Lesion Size (mm)	Physicochemical Properties/Cytotoxicity	Major outcome	Ref.
4NC-beeswax-CONPs	Peritoneal macrophages were infected with *L. amazonensis* (MHOM/BR/1989/166MJO) promastigotes at a 1:10 ratio.	4.98 μM (SI > 10)		CC_50_ => 50 μM.	4NC-beeswax-CONPs has a better SI than free 4NC and reduces the number of intracellular amastigotes by increasing proinflammatory mediators and microbicidal mediators (TNF-α, NO and ROS).	[92]
4NC		4.04 μM (SI = 2.16)		CC_50_ = 8.73 μM		
* BA-KNPs (20)	*L. major* promastigotes.	88%		PS (nm) = 124 ± 14; PDI = 0.3 ± 0.1; ZP (mV) = 6.5 ± 1; LE(%) = 93	BA-KNPs demonstrated significant lethal effects against *L. major* promastigotes and intracellular amastigotes in vitro similar to AmpB-KNPs and greater than free BA.	[6,41]
* BA (20)		~60%		Solubility in water = 0.02 μg/mL		
* Amp-KNPs (20)		89%		PS (nm) = 112 ± 15; PDI = 0.45 ± 0.15; ZP (mV) = 8 ± 1.5; LE (%) = 90		
* BA-KNPs (20)	Peritoneal macrophages infected with *L. major* promastigotes.	~81%				
* BA (20)		~65%				
* AmpB-KNPs (20)		~81%				
Li-ZNPs	*L. major* (MHOM/TM/82/Lev) promastigotes were incubated in the log phase (1 × 10^5^/mL).	53.1		PS (nm) = 105; ZP (mV) = 28.3	Li-ZNPs are more effective than the common drug Glu against CL in mice, reducing lesion size and stimulating cellular immunity through apoptosis and upregulation of proinflammatory cytokines. In addition, they have the potential to reduce oxidative stress and increase antioxidant activity.	[93]
Glu		94.3				
Li-ZNPs	J774-A1 macrophages were infected with *L. major* promastigotes at a 1:10 ratio.	22.6 (SI = 12.35)		CC₅₀ = 279.3 μg/mL		
Glu		30.3 (SI = 16.4)		CC₅₀ = 496.3 μg/mL		
Li-ZNPs-Glu (15)	BALB/c mice were infected with *L. major* promastigotes (SQ., 10^6^ parasites/mL) in the tail.		0			
Li-ZNPs (25)			<5			
Glu (15)			<5			
RsvNE	*L. major* (MRHO/IR/75/ER) promastigotes were incubated in the log phase (10^5^/mL)	35.71		PS (nm) = 110.1 ± 14 nm; PDI = 0.23; ZP (mV) = −48.7	In vivo RsvNE was less effective than free Rsv after 12 h, but in a 21-day study, it reduced lesion size more effectively than the negative control, showing better long-term in vivo efficacy.	[40,94]
Rsv		16.23		Solubility in water (μg/mL) = 21; PS (nm) = 432.7 ± 26; ZP (mV) = −12.1		
RsvNE (20)	BALB/c mice were infected with L. major promastigotes (SQ, 1.6 × 10^6^ parasites/mL) in the tail.		0			
Rsv (20)			<2			
PBS			~7.5			
Cur-AuNPs	*L. major* (MRHO/IR/75/ER) promastigotes were incubated in the stationary phase (2 × 10^6^ parasites/mL).	29.89		PS (nm) = 22.2 ± 12.7 nm	Cur-AuNPs inhibited parasite proliferation less effectively than AmpB but without significant toxicity. In addition, they reduced inflammation in infected mice, achieving similar efficacy to AmpB, in part by inducing a Th1 response.	[95]
AmpB		<1.1		CC₅₀ = 29.80 μg/mL		
Cur-AuNPs	J774-A1 macrophages were infected with L. major promastigotes at a 1:10 ratio.	54.04		Cell viability (%) => 90		
AmpB		2.4				
* Cur-AuNPs (60)	BALB/c mice were infected with promastigotes (SQ).		<0.5			
AmpB (8)			<0.5			
* QT-Phys (400)	*L. major* (MRHO/IR/75/ER) promastigotes were incubated (1 × 10^5^ parasites/mL).	100%		PS (nm) = ~88; EEn (%) = ~96.33	QT-Phys eliminated 100% of the parasites and cured lesions in 28 days with no toxicity.	[96]
* QT-Phys (400)	BALB/c mice were infected with *L. major* promastigotes (i.d., 10^6^ parasites/mL) in the tail.		0	Cell viability (%) = 96.33		

Beeswax-CONPs: beeswax-copaiba oil nanoparticles; 4NC: 4-Nitrochalcone; BA: betulinic acid; KNPs: chitosan nanoparticles; AmpB: amphotericin B; Li: linalool; ZNPs: zinc oxide nanoparticles; Glu: glucantime; NE: nanoemulsion; Rsv: resveratrol; AuNPs: gold nanoparticles; Cur: curcumin; Phys: phytosomes; QT: QT: quercetin; SI: selectivity index; * µg/mL; CC_50_: cytotoxic concentration 50%; PS: particle size; PDI: polydispersity Index; ZP: zeta potential; EEn: encapsulation efficiency; LE: load efficiency; SQ: subcutaneous.

**Table 5 pharmaceutics-16-01239-t005:** Phytochemical-loaded nanosystems for treatment of cutaneous leishmaniasis caused by *L. mexicana* and *L. tropica*.

Nanosystem/Monotherapy (mg/kg)	Model/Parasite	IC_50_ (µg/mL)/Lesion size	Physicochemical Properties/Cytotoxicity	Major Outcome	Ref.
MTF-Apg-SGNTs	*L. mexicana* (MNYC/BZ/62/M379) promastigotes were incubated in the stationary phase (2 × 10^6^ cells/well).	1.4 ± 0.18	VS (nm) = 127 ± 1.2; PDI = 0.163; EE (%) = 76.5 ± 4.1/93.3 ± 3.6	MTF-Apg-SGNTs exhibited higher activity on both promastigotes and intracellular amastigotes than Apg-NTs and free Apg and displayed higher EE.	[120]
Apg-NTs		8.6 ± 0.97	VS (nm) = 254 ± 2.4; PDI = 0.125; EE (%) = 81.6 ± 4.7		
Apg		9.8 ± 0.94			
MTF-Cur-SGNTs		8.7 ± 0.98	VS (nm) = 139 ± 2.8; PDI = 0.193; EE (%) = 79.7 ± 4.3/85.3 ± 3.7		
Cur-NTs		28.7 ± 2.11	VS (nm) = 284 ± 2.2; PDI = 0.132; EE (%) = 85.3 ± 4.5		
Cur		36.3 ± 2.13			
MTF-Ltl-SGNTs		8.4 ± 1.61	VS (nm) = 133 ± 1.5; PDI = 0.154; EE (%) = 77.5 ± 3.7/87.6 ± 3.2		
Ltl-NTs		30.5 ± 2.09	VS (nm) = 267 ± 1.4; PDI = 0.137; EE (%) = 87.1 ± 4.2		
Ltl		32.8 ± 2.35			
MTF-QT-SGNTs		8.4 ± 1.61	VS (nm) = 131 ± 2.1; PDI = 0.186; EE (%) = 77.3 ± 4.1/89.3 ± 4.7		
QT-NTs		35.4 ± 2.87	VS (nm) = 269 ± 1.8; PDI = 0.183; EE (%) = 88.7 ± 4.4		
QT		41.4			
MTF-Apg-SGNTs	Bone marrow-derived macrophages were infected with *L. mexicana* promastigotes at a 1:10 ratio.	0.6 ± 0.08			
Apg-NTs		4.5 ± 0.37			
Apg		6.7 ± 0.44			
MTF-Cur-SGNTs		1.9 ± 0.19			
Cur-NTs		6.8 ± 0.65			
Cur		9.8 ± 0.71			
MTF-Ltl-SGNTs		2.9 ± 0.29			
Ltl-NTs		10.8 ± 1.41			
Ltl		11.5 ± 1.33			
MTF-QT-SGNTs		2.3 ± 0.27			
QT-NTs		10.4 ± 1.71			
QT		13.8 ± 1.57			
NTZ-QT-NTs	*L. tropica* promastigotes were incubated in the stationary phase (1 × 10^6^ parasites/mL).	3.15 ± 0.89	PS (nm) = 210.90 ± 3.67 ; PDI = 0.16 ± 0.009; ZP (mV) = −15.1 ± 1.48; EE (%) = 85 ± 0.02; CC₅₀ = 71.95 ± 3.32 μg/mL	NTZ-QT-NTs demonstrated superior efficacy to QT-NTs and QT dispersion, being ~4.73 and 13.87 times more effective, respectively, in in vitro studies. In mice, the use of NTZ-QT-NTs reduced lesion size ~15 times more than the exclusive use of NTZ-QT-Gel.	[121]
NTZ-NTs		8.35 ± 0.95			
QT-NTs		14.91 ± 1.09			
NTZ-QT dispersion		19.66 ± 1.17	CC₅₀ = 49.77 ± 2.15 μg/mL		
NTZ dispersion		29.67 ± 1.43			
QT dispersion		43.72 ± 1.52			
NTZ-QT-NTS-Gel	BALB/c mice were infected with *L. tropica* promastigotes (SQ) in the right ear.	0.2 ± 0.1 mm	Amount of NTZ and QT permeated = 202.85 μg/cm^2^ and 262.72 μg/cm^2^		
NTZ-QT-Gel		3.1 ± 0.3 mm	Amount of NTZ and QT permeated = 40.54 μg/cm^2^ and 59.37 μg/cm^2^		
Untreated control		7.9 ± 0.6 mm			

MTF: miltefosine; SGNTs: second-generation nano-transfersomes; NTs: nanotransfersomes; Apg: apigenin; Cur: curcumin; Ltl: luteolin; QT: quercetin; NTZ: nitazoxanide; VS: vesicle size; EE: entrapment efficiency; PS: particle size; PDI: polydispersity index; ZP: zeta potential; CC_50_: cytotoxic concentration 50%; SQ: subcutaneous.

**Table 6 pharmaceutics-16-01239-t006:** Phytochemical combination therapy for the treatment of trypanosomiasis and schistosomiasis.

Combination therapy/Monotherapy (μM)	Parasite	EC_50_ (µM)/Dead (%)	Major Outcomes	Ref.
CD24-Cur	*T.b.brucei* 449 cell line	4.85 ± 0.02 (SI > 14.4)	CD24-Cur demonstrated superior antitrypanosomal activity compared to free CD24, along with greater selectivity.	[132]
Cur		3.12 ± 0.43 (SI > 22.4)		
CD24		10.1 ± 0.5 (SI > 6.9)		
RK-52-Cur		4.64 ± 0.35	RK-52-Cur did not result in a significant enhancement in activity compared to the free RK-52.	[133]
Cur		3.12 ± 0.43		
RK-52		2.33 ± 0.29		
PZQ-Rsv (100)	*S. mansoni* NTS were incubated (50–100 parasites/well)	81.0 ± 5.2	The combination of PZQ or As with Rsv enhanced the antischistosomal effect by 60% and 30%, respectively, compared to a single treatment.	[134]
Rsv (100)		30.0 ± 1.6		
PZQ (100)		56.9 ± 2.5		
As-Rsv (100)		99.9 ± 0.1		
As (100)		70.0 ± 3.8		

Rsv: resveratrol; Cur: curcumin; PZQ: praziquantel; As artesunate; NTS: newly transformed schistosomula; SI: selectivity index; EC₅₀: half maximal effective concentration.

**Table 7 pharmaceutics-16-01239-t007:** Phytochemical-loaded nanosystems to treat schistosomiasis.

Nanosystem/Monotherapy (mg/kg)	Model	Burden Worms (%)	Egg Burden (%)	Physicochemical Properties/Biochemical Parameters/Cytotoxicity	Major Outcomes	Ref.
CvNE (200)	Mice were infected with *S. mansoni* cercariae (SQ, 80 parasites)—21 days post-infection (immature stage).	86.4	Feces = 90.1	PS (nm) = 124 ± 0.8; PDI = 0.20 ± 0.01; ZP (mV) = −26.4 ± 0.59; cell viability (%) =≥ 95	Juvenile parasites appeared to be more sensitive to CVNE than adult stages, as a reduction of 86.6 and 54.7% in total worm burden was observed in prepatent and patent infection, respectively.	[158]
Cv (200)		30.3	Feces = ~30			
PZQ (400)		29.2	Feces = 31.9			
CvNE (200)	Mice were infected with *S. mansoni* cercariae (SQ, 80 parasites)—42 days post-infection (adult stage)	54.8	Feces = 50.1			
Cv (200)		<54.8	Feces =< 50.1			
PZQ (400)		84.2	Feces = 90.9			
Lico A-SLNs (5)	Swiss mice were infected with *S. mansoni* cercariae (i.p., 70 parasites)—49–53 days post-infection (adult stage)	52.9	Intestinal: immature = 50.8; mature = 38	PS (nm) = 101.1 ± 1.8; PDI = 0.17 ± 0.01; ZP (mV) = −32.34 ± 2.9; EE (%) = 98.33; CC50 (μM) = 956.2.	L-SLNs improved in vivo antischistosomal effects compared with free Lico A. L-SLNs possess lower cytotoxic compared with free Lico A.	[166]
Lico A (5)		51.2	Intestinal: Immature = 45.5; Mature =< 38	CC₅₀ (μM) = 20.7; oral bioavailability (%) = 3.3		
Cur- AuNPs (400)	C57BL/6 mice were infected with *S. mansoni* cercariae (SQ, 120 ± 10 parasites)—21–35 days post-infection	42.42	Intestinal = 77.26; hepatic = 83.85	AST (IU/L) = 31.75 ± 1.28; ALT (IU/L) = 28.88 ± 1.55	Cur-AuNPs-PZQ more efficiently reduced worm and egg burdens in the intestine and liver of mice than Cur monotherapy and were also less toxic than PZQ	[167]
Cur (400)		45.45	Intestinal = 37.05; hepatic = 34.12	AST (IU/L) = 46.75 ± 5.26; ALT (IU/L) = 46.25 ± 3.62		
Cur-AuNPs-PZQ (400)		97.43	Intestinal = 73.77; hepatic = 46.67	AST (IU/L) = 31 ± 1.85; ALT (IU/L) = 30.5 ± 2.08		
Cur-PZQ (400)		73.75	Intestinal = 42.95; hepatic = 54.92	AST (IU/L) = 47.13 ± 3.72; ALT (IU/L) = 41.13 ± 2.42		
PZQ (500)		72.72	Intestinal = 92.98; hepatic = 93.60.	AST (IU/L) = 51.63 ± 2.72; ALT (IU/L) = 47.13 ± 4.22		
Lic-A-PCLs (20)	BALB/c mice were infected with *S. mansoni* LE cercariae (80 parasites) in the tail—49 days post-infection (adult stage)	56.3	Hepatic = 27.1; spleen = 95.	PS (nm) = 131.4; ZP (mV) = −39.9	Lic-A-PCLs reduced the blood worm burden similarly to free Lic-A but to a lesser extent than PZQ.	[168]
Lic-A-PCLs (200)		41.7	Hepatic = 25.21; Spleen = 65	PS (nm) = 233.2; ZP (mV) = −39.9		
Lic-A (200)		44.2	Hepatic = 42; spleen = 96.6			
PZQ (400)		85	Hepatic = 24.21; spleen = 92.7.			

Cv: carvacrol; NE: nanoemulsion; PZQ: praziquantel: Lic-A: licochalcone A; SLNs: solid lipid nanoparticles; Cur: curcumin, AuNPs: gold nanoparticles; Lic-A: (±)-licarin-A: PCLs: poly-Ɛ-caprolactone nanoparticles; PS: particle size; PDI: polydispersity index; ZP: zeta potential; EE: entrapment efficiency; CC_50_: cytotoxic concentration 50%; AST: aspartate aminotransferase; ALT: Alanine aminotransferase; SQ: Subcutaneous; i.p.: intraperitoneal.

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
