# Peer review of "Combination Therapy and Phytochemical-Loaded Nanosytems for the Treatment of Neglected Tropical Diseases"

_pharmaceutics, 2024, doi:10.3390/pharmaceutics16101239_

Round 1
Reviewer 1 Report
Comments and Suggestions for Authors
In this manuscript authors have review the impact of combination therapy with phytochemicals for the treatment of neglected tropical diseases (NTDs), including leishmaniasis, trypanosomiasis, and schitosomiasis. The manuscript is well written, however minor modifications are required:
(i) In Figure 1 and Figure 2, legend the full names of the chemical should be provided.
(ii) In tables, an additional column of the major outcome should be inserted.
(iii) In some table “model” is written as “modelo”
(iv) The manuscript should include some mechanistic figures which should describe the impact of combination therapy for these diseases.
Comments on the Quality of English LanguageIt's fine
Author Response
Dear Reviewer,
Thank you very much for taking the time to review manuscript ID: pharmaceutics-3171001. We greatly appreciate your valuable comments and suggestions. Below, you will find our detailed responses and the corresponding corrections highlighted in green in the resubmitted files.
Comment 1: In Figure 1 and Figure 2, the full names of the chemicals should be provided in the legend.
Response: Thank you for pointing this out. We agree with your suggestion. The legends for Figure 1 and Figure 2 have been updated to include the full names of the chemicals. These changes are highlighted in green in the figures.
Comment 2: In tables, an additional column for the major outcome should be inserted.
Response: We appreciate this suggestion. We have inserted an additional column in the tables to include the major outcomes, as requested. The changes are highlighted in green in the revised tables.
Comment 3: In some tables, “model” is written as “modelo.”
Response: Thank you for identifying this inconsistency. We have corrected the term "modelo" to "model" in the relevant tables. The corrections are highlighted in green.
Comment 4: The manuscript should include some mechanistic figures that describe the impact of combination therapy for these diseases.
Response: We appreciate this recommendation. We have added a figure to the manuscript to illustrate. These figure is included in the revised manuscript and are highlighted in green.
Once again, we sincerely thank you for your valuable feedback and suggestions, which have been instrumental in enhancing the quality of our manuscript. We remain at your disposal for any further comments or questions.
Reviewer 2 Report
Comments and Suggestions for Authors
The manuscript is interesting, however, some improvements are required prior publication.
Thus, the manuscript includes many abbreviations of the compounds, which are not adequately presented in the text, which, in my opinion, makes it difficult for readers who are not specialized in this field to understand it.
Moreover, some abbreviations are presented as a legend to the tables, but subsequently only abbreviations are used in the text, which makes it difficult to follow in the text.
- line 176 write the chemical name of LUP and UA before abbreviation
- line 180 include few details on the types of nanostructured lipid carriers180
(NLCs) with specific action
- line 326 include the name of BA before abbreviation
- line 335 include the name of Li-ZNPs before abbreviation
- line 351 include the name of Rsv before abbreviation
- line 355 include the name of RsvNE,before abbreviation and some few information about it
- line 356 include the name of QT-loaded phytosomes (QT-Phys) before abbreviation and some few information about it
- line 398 include the name of polyphenols (Apg, Cur, Ltl, and QT) loaded in NTs before abbreviation and some few information about it
- line 402 include the name of MTF-Apg,before abbreviation and some few information about it
- line 603 include the name of LicoA,before abbreviation and some few information about it
- line 607 nclude the name of L-SLNs before abbreviation and some few information about it
same for Lico A-SLNs, Cur-PZQ, Lic-A-PCLs, and so on
Author Response
Dear Reviewer,
Thank you very much for taking the time to review manuscript ID: pharmaceutics-3171001. We deeply appreciate your valuable comments and suggestions. Below, you will find our detailed responses and the corresponding corrections highlighted in red in the resubmitted files.
Comment 1: line 176 write the chemical name of LUP and UA before abbreviation
Response: Thank you for pointing this out. We agree with this comment and have made the suggested change. The full names of LUP and UA are now included before the abbreviations in line 179 (see line 179, highlighted in red).
Comment 2: line 180 include few details on the types of nanostructured lipid carriers180
(NLCs) with specific action
Response: We appreciate this suggestion. We have added details about the types of NLCs and their specific actions in line 184. This additional information is highlighted in the text (see line 184, highlighted in red).
Comment 3: line 326 include the name of BA before abbreviation
Response: We have included the full name of BA before the abbreviation in line 326, as requested (see line 338, highlighted in red).
Comment 4: line 335 include the name of Li-ZNPs before abbreviation
Response: The full name of Li-ZNPs has been added before the abbreviation in line 335 (see line 348, highlighted in red).
Comment 5: line 351 include the name of Rsv before abbreviation
Response: The full name of Rsv is now included before the abbreviation in line 351 (see line 364, highlighted in red).
Comment 6: line 355 include the name of RsvNE,before abbreviation and some few information about it.
Response: We have incorporated the full name of RsvNE before the abbreviation and added additional information about it in line 368 (see line 368, highlighted in red).
Comment 7: line 356 include the name of QT-loaded phytosomes (QT-Phys) before abbreviation and some few information about it
Response: The full name of QT-loaded phytosomes (QT-Phys) and the requested additional information have been added in line 376 (see line 376, highlighted in red).
Comment 8: line 398 include the name of polyphenols (Apg, Cur, Ltl, and QT) loaded in NTs before abbreviation and some few information about it
Response: We have included the full names of the polyphenols (Apg, Cur, Ltl, and QT) and their loading in NTs before the abbreviations in line 419, along with additional information (see line 419, highlighted in red).
Comment 9: line 402 include the name of MTF-Apg,before abbreviation and some few information about it
Response: The full name of MTF-Apg and the requested additional information have been added in line 423 (see line 423, highlighted in red).
Comment 10: line 603 include the name of LicoA,before abbreviation and some few information about it .
Response: We have included the full name of LicoA before the abbreviation and added additional information in line 603 (see line 603, highlighted in red).
Comment 11: Line 607: Include the name of L-SLNs before abbreviation and some few information about it same for Lico A-SLNs, Cur-PZQ, Lic-A-PCLs, and so on
Response: The full name of L-SLNs and the requested additional information have been included in line 607, as well as for Lico A-SLNs, Cur-PZQ, Lic-A-PCLs, and other terms mentioned (see lines 634,636,657,667,160,167,197,204,268,313,461,464,465,506,519,528,532,604,622,617 highlighted in red).
Once again, we appreciate your valuable feedback and suggestions, which have significantly contributed to the improvement of our manuscript. We are at your disposal for any further comments or questions.
Reviewer 3 Report
Comments and Suggestions for Authors
Dear Authors,
The manuscript of Jacqueline Soto-Sánchez and Gilberto Garza-Treviño, entitled “ Combination Therapy and Phytochemical-Loaded Nanosytems for the Treatment of Neglected Tropical Diseases” is both interesting and relevant at the same time, trying to delineate the main contributions to combination therapy with phytochemicals and nanotechnologies over the past five years, with a focus on the delivery of secondary metabolites.
There are some issues that need to be addressed during the review and are listed below:
1. The bibliography should be organized in ascending order of numbers.
2. In the case of Table 1: a more detailed description of the mechanism of action of tropical antiparasitic medication must be done, highlighting both advantages and disadvantages in a distinctive manner depending on the tropical parasitic agent involved.
3. A table (similar to Table 1) should be created to list natural compounds or alternative medications that have been used in therapy or have clinical potential for tropical parasitic agents.
4. In the case of Table 2, it is necessary to provide a more comprehensive explanation of the model used in vitro/in vivo and the pharmacokinetic and pharmacotoxicological parameters. The same process is necessary for the other tables (Table 3, Table 4).
5. The authors can simplify the literature by reducing the order of the subchapters.
6. I also recommend creating a section of future trends that may be beneficial for this study.
Comments on the Quality of English Language
Minor editing of English language required.
Author Response
Dear Reviewer,
Thank you very much for taking the time to review manuscript ID: pharmaceutics-3171001. We greatly appreciate your valuable comments and suggestions. Below, you will find our detailed responses and the corresponding corrections highlighted in blue in the resubmitted files.
Comment 1: The bibliography should be organized in ascending order of numbers.
Response: Thank you for highlighting this issue. We have reorganized the bibliography in ascending numerical order as requested. We have also considered including the tables to further enhance the manuscript. Additional changes were also made to the References.
Comment 2: In Table 1, a more detailed description of the mechanism of action of tropical antiparasitic medication must be provided, highlighting both advantages and disadvantages in a distinctive manner depending on the tropical parasitic agent involved.
Response: We appreciate this suggestion. We have revised Table 1 to include a more detailed description of the mechanism of action of tropical antiparasitic medications, clearly outlining the advantages and disadvantages based on the specific tropical parasitic agents involved. The updated table is highlighted in blue.
Comment 3: A table (similar to Table 1) should be created to list natural compounds or alternative medications that have been used in therapy or have clinical potential for tropical parasitic agents.
Response: Thank you for your suggestion. We have already detailed the information on natural compounds and alternative medications in our previous publications. Given that this information is extensively covered elsewhere, we do not consider it necessary to include a separate table in the current manuscript. However, we appreciate your observation. (https://doi.org/10.2174/1389557522666220404090429; https://doi.org/10.1111/cbdd.14470)
Comment 4: In Table 2, it is necessary to provide a more comprehensive explanation of the model used in vitro/in vivo and the pharmacokinetic and pharmacotoxicological parameters. The same process is necessary for the other tables (Table 3, Table 4).
Response: Thank you for your insightful comment. We have provided a more comprehensive explanation of the models used for Tables 2, 3, 4, 5, 6, and 7. These changes are highlighted in blue in the respective tables.
Regarding pharmacokinetic parameters, the reviewed studies do not include this information as, although some were conducted in mice, this specific data is not available. Instead, we have included the physicochemical properties of the nanoformulations, which we believe will enhance the manuscript. For pharmacotoxicological parameters, we have included cytotoxicity, as this is the most commonly studied parameter in the reviewed articles. These changes are highlighted in blue in the respective tables.
We hope these revisions address your concerns and improve the manuscript.
Comment 5: The authors can simplify the literature by reducing the order of the subchapters.
Response: Thank you for your suggestion. We have simplified the literature sections 2 and 2.1 by reducing the order of the subchapters to enhance clarity and coherence. These changes are highlighted in blue in the revised manuscript. Given the nature of the information, we have decided to retain the original order to ensure greater clarity and completeness. We hope this approach meets your expectations and improves the manuscript.
Comment 6: I also recommend creating a section on future trends that may be beneficial for this study.
Response: We appreciate this recommendation. A new section on future trends relevant to the study has been added to the manuscript. This section is highlighted in blue.
Once again, we sincerely thank you for your insightful feedback and suggestions, which have significantly contributed to improving the quality of our manuscript. We remain at your disposal for any further comments or questions.
Round 2
Reviewer 2 Report
Comments and Suggestions for Authors
Agree with the revised manuscript.